# Applying Machine Learning to Healthcare Operations Management: CNN-Based Model for Malaria Diagnosis

**DOI:** 10.3390/healthcare11121779

**Published:** 2023-06-16

**Authors:** Young Sik Cho, Paul C. Hong

**Affiliations:** 1College of Business, Jackson State University, Jackson, MS 39217, USA; 2John B. and Lillian E. Neff College of Business and Innovation, The University of Toledo, Toledo, OH 43606, USA

**Keywords:** machine learning, convolutional neural networks, healthcare operations management, epidemic diagnosis, malaria, global supply chain disruption, operational capabilities, healthcare performance, k-fold cross-validation test, artificial intelligence

## Abstract

The purpose of this study is to explore how machine learning technologies can improve healthcare operations management. A machine learning-based model to solve a specific medical problem is developed to achieve this research purpose. Specifically, this study presents an AI solution for malaria infection diagnosis by applying the CNN (convolutional neural network) algorithm. Based on malaria microscopy image data from the NIH National Library of Medicine, a total of 24,958 images were used for deep learning training, and 2600 images were selected for final testing of the proposed diagnostic architecture. The empirical results indicate that the CNN diagnostic model correctly classified most malaria-infected and non-infected cases with minimal misclassification, with performance metrics of precision (0.97), recall (0.99), and f1-score (0.98) for uninfected cells, and precision (0.99), recall (0.97), and f1-score (0.98) for parasite cells. The CNN diagnostic solution rapidly processed a large number of cases with a high reliable accuracy of 97.81%. The performance of this CNN model was further validated through the k-fold cross-validation test. These results suggest the advantage of machine learning-based diagnostic methods over conventional manual diagnostic methods in improving healthcare operational capabilities in terms of diagnostic quality, processing costs, lead time, and productivity. In addition, a machine learning diagnosis system is more likely to enhance the financial profitability of healthcare operations by reducing the risk of unnecessary medical disputes related to diagnostic errors. As an extension for future research, propositions with a research framework are presented to examine the impacts of machine learning on healthcare operations management for safety and quality of life in global communities.

## 1. Introduction

According to the World Health Organization (WHO), malaria case rates (i.e., cases per 1000 population) fell from 82 in 2000 to 57 in 2019 but rose to 59 in 2020. The WHO reported that this unusual 2020 increase in malaria case rates was related to service supply disruptions during the COVID-19 pandemic [1]. In fact, the number of malaria cases increased from 227 million in 2019 to 241 million in 2020, and the number of malaria deaths in 2020 was estimated at 627,000, a 12% increase from 2019 [2].

Moreover, in the case of malaria, the more severe problem is that the existing malaria diagnosis method relies on direct human observation, which takes much time for diagnosis, making it difficult to test many patients simultaneously. Additionally, there is a limitation in that diagnostic accuracy is greatly affected by variability between observers. In other words, the effectiveness of the conventional microscopic diagnosis is highly dependent on the expertise of parasitologists. Besides, it is common for parasitologists to work in resource-constrained environments without stringent systems to maintain their know-how or diagnostic quality [3]. This can often lead to erroneous diagnoses and inappropriate treatment, which can have fatal consequences [3,4].

As one solution to overcome the problems of the current manual diagnosis of malaria, an automatic classification technique using machine learning (ML) is expected to enable faster and more accurate diagnosis than manual classification. In particular, among several machine learning techniques, convolutional neural networks (CNNs) are widely recognized as the best architecture for image recognition, classification, and detection tasks [5]. Therefore, this study presents an artificial intelligence (AI) solution for malaria diagnosis using CNN algorithms. Then, based on the findings, this study provides an in-depth discussion of how machine learning technologies can improve the operational capabilities, especially in terms of healthcare operations management. In particular, this study seeks to answer the following research questions: (i) Can a CNN-based ML model classify red blood cell images into malaria-infected and malaria-uninfected cells? (ii) How accurately can the CNN-based ML model classify malaria-infected and malaria-uninfected cells? (iii) How can CNN-based ML solutions help improve healthcare operations management?

To answer these research questions, this study has the following structure. We first briefly review previous work in ML research on epidemic detection, then discuss CNN technologies and image data processing as the fundamental methodology in this study. Next, we apply and evaluate different techniques, including batch normalization, data augmentation, and pre-trained models, to develop the best CNN solution design for classifying malaria-infected and non-infected cell types. Based on the study results, practical implications and lessons learned are discussed in the context of healthcare operations management. Finally, we present a benchmarking research framework and proposals to extend the ML approach of this study to other epidemiological situations.

Previous research on the impact of machine learning on healthcare operations management has been mostly an abstract discussion. However, this study empirically applies machine learning techniques to solve specific healthcare problems. In short, this study combines computer science and operations management to provide more practical implications for leveraging machine learning techniques in healthcare operations.

## 2. Literature Review

There are several promising prior studies on the capabilities of ML-based techniques in detecting infectious diseases. For instance, using a machine learning framework, Colubri et al. [6] introduced an application that can predict the outcome of Ebola patients from early clinical symptoms. Smith and Kirby [7] described ML applications for analyzing different types of microbial image data, particularly progress in smear and plate interpretation.

Another notable study on ML-based infectious disease diagnosis is that of Das et al. [8], who developed a computer-aided malaria parasite characterization and classification based on light microscopy images of peripheral blood smears collected from 600 patients using an ML approach. Their proposed ML scheme applying the Bayesian approach provides 84.0% accuracy and 98.1% sensitivity by selecting the 19 most significant features, and the support vector machine (SVM) achieved 83.5% screening accuracy and 96.6% sensitivity with the 9 most significant features [8].

Similarly, there are other studies that have applied various machine learning methods to detect malaria parasites. Bibin et al. [9] proposed a deep belief network (DBN)-based trained model to classify 4100 peripheral blood smear images into parasitic or nonparasitic classes. The proposed method showed an F-score of 89.66%, a sensitivity of 97.60%, and a specificity of 95.92% [9]. Gopakumar et al. [10] used a customized CNN model operating on a focus stack of images for automated quantitative detection of Plasmodium falciparum malaria in blood smears. The detection accuracy of the CNN model was 97.06% sensitivity and 98.50% specificity [10].

Yang et al. [3] developed a method using a deep learning algorithm to detect malaria parasites in thick blood smear images, run on a smartphone. They trained and tested a deep learning method using 1819 thick smear images from 150 patients [3]. The study results showed the effectiveness of the CNN model in distinguishing positive (parasitic) image patches from negative image patches, with performance metrics of accuracy (93.46% ± 0.32%), precision (94.25% ± 1.13%), and negative predictive value (92.74% ± 1.09%) [3].

Especially in the case of the COVID-19 pandemic, Dandekar et al. [11] applied the neural network module of ML to develop a globally applicable COVID-19 diagnosis model to analyze and compare the role of quarantine control policies globally across the continents of Europe, North America, South America, and Asia. Dandekar et al. [11] also hosted quarantine diagnosis results from 70 countries around the world on a public platform: https://covid19ml.org/ (accessed on 15 March 2023).

One example of a notable literature review source for ML-based infectious disease diagnosis is the work of Baldominos et al. [12]. The study performed a computer-based systematic literature review in order to investigate where and how computational intelligence (i.e., different types of machine learning techniques) is being utilized to predict patient infection [12].

Deep learning, a specific subset of machine learning, is a computational processing system composed of artificial neural networks, heavily inspired by how biological nervous systems process information and make decisions [13]. Deep learning allows for incrementally learning complex input data features by going through the architecture’s hidden layers [14]. That is, as the input data pass through hidden layers, the complexity of the input data is computed as a simpler and less abstract concept for the final output, which is the so-called nested hierarchical approach [14,15,16].

For more information on deep learning, see the work by Alzubaidi et al. [17], which presents essential overall aspects of deep learning and provides a clear image of deep learning in one review paper. In our study, CNN was adopted among various deep learning techniques, and a detailed description of CNN follows in the next section.

## 3. Methodology

### 3.1. Convolutional Neural Networks

Traditional artificial neural networks (ANNs) do not have a good understanding of local representations or patterns of image data. However, convolutional neural networks (CNNs) have a filter that goes through the entire image, so CNNs can perform convolutional operations to detect basic information available in the image, such as edges or curves. In other words, CNNs can learn local spatiality within images and use filters to detect features across images. As a result, CNNs allow image-related features to be encoded into the architecture, making the network more suitable for image-centric tasks [13]. In short, CNNs have advantages over traditional ANNs in image data prediction, including spatial and transform invariance and extracting essential features. Accordingly, CNN is expected to be a more practical solution design for the goal of this study, which is to classify images of malaria-infected cells.

The basic architecture of CNNs is illustrated in Figure 1 using the example of malaria diagnosis. CNNs typically consist of three types of layers: convolutional, pooling, and fully connected [13]. The convolutional layers extract features from an input image by finding the spatial locality of the image by sliding a patch-sized local detector across the image [15]. The pooling layers are then used between subsequent convolutional layers to reduce the number of trainable parameters by retaining image information in a reduced spatial size [18]. Lastly, the fully connected layers classify images based on the features extracted through the previous layers and their different filters [19]. While convolutional and pooling layers tend to use the ReLu function, the fully connected layers usually use the SoftMax activation function to classify the input to generate probabilities between 0 and 1 [19].

In summary, earlier layers focus on superficial features such as color and edges, but as the image data progress through the CNN’s layers, it begins to recognize more prominent elements or shapes of objects until it finally identifies the intended object [19].

### 3.2. Data Sampling

This study was conducted using malaria microscopic image data (n = 27,558) of red blood cells (RBC) provided by the MIT Applied Data Science Program in 2022. However, these malaria data are open source and published by the National Library of Medicine at the National Institutes of Health [20]. The complete dataset can be downloaded from the National Library of Medicine–Malaria Data [21]. The malaria data are thin smears already classified into the following two categories: (i) parasitized—infected cells, including Plasmodium parasites that cause malaria, and (ii) uninfected—uninfected cells free of Plasmodium parasites. The data were randomly divided into 24,958 training images and 2600 test images.

### 3.3. Image Data Processing

Before building the CNN model and evaluating its performance, the necessary pre-processing was performed as follows. All images were equally sized and transformed into a 2D array for use as input to the CNN model. Then, the train and test images were normalized, and labels were created for both types of images. Figure 2 shows the visualized sample images from the training data. Figure 3 presents averaged images of parasitic and uninfected erythrocytes. Finally, the training and test data images were processed using Gaussian blurring. Figure 4 presents the results of applying Gaussian blurring to images from the training data (label 1 = parasitized, label 0 = uninfected). The coding details of image processing using Gaussian blurring are provided in Appendix A.

## 4. Analysis

### 4.1. Test Results

First, a base model, which contained a convolutional layer with 32 filters, was built to evaluate the performance of the CNN architecture. Then, other updated CNN models were evaluated to find the best solution design to classify malaria-infected and uninfected cell types, including batch normalization, data augmentation, and a pre-trained model (VGG16). Table 1 below summarizes the updated techniques and performances for each CNN model.

As a result of the test, Model 2, which added a second convolutional layer with 64 filters and performed batch normalization (i.e., normalizing inputs of each layer), showed the highest model accuracy at 0.9781, while Model 3, using the data augmentation technique, demonstrated the lowest model accuracy, with 0.5000. Figure 5 shows an augmented image visualization of RDCs in Model 3. Then, Figure 6 represents training (i.e., training data) and validation (i.e., test data) accuracy plots for Models 1, 2, 3, 4, and 5, respectively. As can be seen from the accuracy plots, Model 2 exhibited the highest accuracy improvement on both training and test data with increasing epochs, so it is considered the most reliable CNN architecture for malaria detection.

### 4.2. Solution Model

Additional in-depth evaluation of the CNN models developed in this study was performed with the Confusion Matrix’s precision, recall, f1-score, and accuracy. Here, precision indicates the proportion of correct predictions in predictions of the positive class, while recall measures the proportion of correct predictions in the positive class [22]. The f1-score represents the weighted harmonic mean of precision and recall, while accuracy measures the proportion of correct predictions within a model [19]. The formula is presented in Equations (1)–(4). TP = true positive, TN = true negative, FP = false positive, and FN = false negative are the four main parameters of the confusion matrix [23].
(1)$$\mathrm{Precision}=\frac{Number\;of\;TP}{Number\;of\;(TP+FP)}$$
(2)$$\mathrm{Recall}=\frac{Number\;of\;TP}{Number\;of\;(TP+FN)}$$
(3)$$F1-\mathrm{score}=2\;*\;\frac{Precision\;*\;Recall}{Precision+Recall}$$
(4)$$\mathrm{Accuracy}=\frac{Number\;of\;(TP+TN)}{Number\;of\;(TP+TN+FP+FN)}$$

Figure 7 shows the confusion matrix for Model 2 tested with 2600 test images. Label 1 represents parasitic cells, and label 0 illustrates uninfected cells, as classified in the data processing. The test results were: precision = 0.97, recall = 0.99, and f1-score = 0.98 for uninfected cells (n = 1300), and precision = 0.99, recall = 0.97, and f1-score = 0.98 for parasite cells (n = 1300). In summary, as shown in Figure 7, Model 2 correctly classified most malaria-infected and non-infected cases, with minimal misclassification. Again, compared to the other CNN models tested in this study, Model 2 most accurately predicted and classified RBCs as parasitized versus uninfected (accuracy = 0.98) using the 2600 test image data. Based on these results, Model 2 was determined to be the best alternative for malaria detection in this given dataset. The coding details of Model 2, as a final solution model, are provided in Appendix B.

### 4.3. Overfitting Test

Finally, we performed k-fold cross-validation to assess whether solution Model 2 was overfitting. To address the overfitting problem in machine learning, we generated a total of 4 different validation subsets (n = 2600) consisting of 1300 parasitic and 1300 uninfected images. The machine learning algorithm developed in this study was then iteratively trained with those four different fold datasets and tested using the remaining four split-validation subset data, as shown in Figure 8.

The result showed an average model accuracy of 0.9736, with a minimum of 0.9700 (k = 2-fold) and a maximum of 0.9781 (k = 1-fold), as presented in Figure 8. Further, test results from each validation subset showed similar precision, recall, and f1-scores, correctly classifying most malaria-infected and non-infected cases, with minimal misclassification, as reported in Table 2. Taken together, it was concluded that the threat of overfitting is not a major problem for the validity of the machine learning model developed in this study.

## 5. Discussion

### 5.1. Implications for Healthcare Operations

The test results of this study provided clear answers to our research questions 1 and 2 by providing empirical evidence that CNNs can be a valuable solution for rapidly and accurately diagnosing large numbers of malaria cases. Based on the findings of this study, we now discuss research question 3 on how CNN-based ML solutions can help improve healthcare operations management.

According to the 2021 WHO world malaria report [2], 29 countries accounted for 96% of global malaria cases, including Nigeria (27%), Congo (12%), Uganda (5%), Mozambique (4%), and Angola (3.4%). The Africa region accounted for approximately 95% of malaria cases in 2020. As Figure 9 shows, majority of malaria cases occurred in underdeveloped countries between 2000 and 2020.

These underdeveloped countries are likely to lack medical professionals and healthcare facilities for diagnosing malaria. This study empirically demonstrates that large-volume malaria diagnoses can be automated with high confidence using ML solutions. Consequently, ML-based automated disease detection systems are expected to have clear advantages for healthcare operations, especially in underdeveloped countries. This means that ML-based systems can rapidly diagnose large numbers of disease cases at a fraction of the cost of expensive medical professionals who can perform these diagnoses.

Next, this study demonstrated that the CNN-based ML solution excels in disease image prediction, especially in malaria diagnosis, with up to 97.81% accuracy. In a similar vein, Rahman et al. [24] studied pneumonia diagnosis using the CNN algorithm, and the results showed 98% classification accuracy between normal and pneumonia images, 95% accuracy between bacterial and viral pneumonia images, and 93% accuracy between normal, bacterial, and viral pneumonia. Spanhol et al. [25] also showed that CNN performance in breast cancer detection is better than previously reported results from other machine learning techniques. This empirical evidence supports that CNN-based diagnostic solutions can be utilized as an alternative solution to address the accuracy variance of manual disease diagnosis discussed earlier in this study.

In addition, the CNN-based diagnosis solution is expected to be used as a secondary result that can be cross-validated by medical staff for various other medical diagnoses, such as tumors and dermatological diseases, beyond malaria. Subsequently, this cross-validation through CNN is expected to improve healthcare operations by not only supporting medical staff’s decision-making but also lowering the risk of unnecessary medical disputes due to misdiagnosis.

### 5.2. Propositions for Healthcare Operations

Researchers of diverse disciplines appreciate machine learning approaches and methods to innovative value creation and delivery [26,27,28]. As an extension of this research, this section presents several propositions for ML solutions in the context of healthcare operations management. Figure 10 shows a proposed research framework of ML-based diagnostic systems to improve healthcare productivity and operational effectiveness. The context assumes global supply chain disruptions related to massive healthcare needs comparable to the COVID-19 pandemic.

Global supply chain disruptions occur when a pandemic such as COVID-19 hits worldwide. As healthcare facilities struggle with surging patients, the constraints of healthcare resources may soon become a bottleneck that may slow down the whole healthcare process, even to the level of a dysfunctional state. This is all the more acute in emerging economies (e.g., Africa, South Asia, Central, and Latin America) with limited healthcare resources [29,30,31]. Based on these empirical case study findings, it is worth comparing the usage of a manual diagnosis system (existing actual method) and an ML-based diagnosis system (proposed method) in terms of healthcare operations. Thus, we propose:

**P1.** *In the case of a global supply chain disruption, healthcare facilities with mostly manual diagnostic systems are more likely to experience healthcare bottlenecks sooner than healthcare facilities with machine learning-based diagnostic systems*.

Appropriate assessment by healthcare operating processes provides insight into quantifiable healthcare capabilities [32,33]. It can help evaluate the criteria for justifying an investment in an ML-based diagnostic system. Moreover, these benchmarking comparisons allow healthcare institutions to prioritize needs based on time, cost, productivity, capacity, and diagnostic quality reliability [34,35,36]. Thus, we propose:

**P2.** *Healthcare facilities that use machine learning-based diagnostic systems are more likely to report better healthcare operational capabilities (e.g., lead times, labor costs, manpower productivity, capacity, accuracy, and other operating indicators) than those that mostly rely on manual diagnostic systems*.

These healthcare operations capability indicators can be used for further backend outcomes of healthcare performance metrics, such as patient and healthcare staff satisfaction ratings, actual determinations of revenues and costs, and other crucial healthcare treatment outcomes such as fatality and cure rates [32,37,38]. Thus, we propose:

**P3.** *Healthcare facilities that use machine learning-based diagnostic systems are more likely to report better healthcare performance outcomes (e.g., patient and staff satisfaction, fatality and cure rates, and other financial ratios) than those that rely mostly on manual diagnostic systems*.

### 5.3. Research Limitations and Suggestions

Despite the advantages of these CNN-based diagnostic systems, caution is needed before readily adopting CNN as a generalizable medical diagnosis method.

First, the CNN architecture is susceptible to small image perturbations. Consequently, if multiple disease factors are included in a single case image, this customized CNN model may not be suitable for precise disease diagnosis. Second, applying CNN solutions requires timely access to large-scale and high-quality image data. Systematic collection and management of medical diagnosis data are crucial to maintaining process integrity for reliable CNN solutions.

Therefore, credible health organizations are expected to publicly disclose infectious disease-related biomedical data immediately from the onset of the pandemic. It is imperative that responsible governance systems are organized at the national level [39,40]. International organizations such as WHO and the UN need to be involved in monitoring, collecting, and sharing biomedical data using effective health information systems [40]. Such a global healthcare ecosystem architecture will enable the rapid development of customized ML-based diagnosis systems in case of the next rounds of pandemic outbreaks.

### 5.4. Future Research Direction

This study has highlighted that machine learning-based AI solutions can replace manual diagnosis methods of diseases (especially malaria). Such machine learning-based AI solutions can bring innovative results to future healthcare systems. In fact, machine learning (ML) has increased its practical applications in various areas of our daily lives. We watch AI-embedded videos, purchase products with AI-enabled advertisements, and even invest in AI-recommended stocks. In particular, with the advent of ChatGPT and other alternatives, we have entered an era of Generative AI systems.

It is known that the number of neurons in the human brain is about 100 billion, and the number of synapses is about 100 trillion. However, the number of parameters of the upcoming GPT 4 or 5 is expected to be about 100 trillion, which is expected to overtake the number of synapses in the human brain in the near future [41]. In other words, the ‘technological singularity’, a hypothetical future point in time when technological growth becomes uncontrollable and irreversible and brings unpredictable changes to human civilization, might not be too far away [42,43]. Therefore, considering the tremendous speed and power of ML, there is an urgent call for research on how to establish international norms and ethical standards to control and manage ML technologies before it goes beyond the singularity [44,45,46].

In addition, to establish more practical norms and ethics for these ML technologies, it is imperative to utilize consortiums composed of not only computer scientists but also experts in various fields, such as medicine, culture, education, government, military, media, and business, to which these ML technologies will be applied [47,48,49]. In summary, ML technology is expected to expand the research frontiers that require a greater level of international collaborative efforts and research to capture the potential of ML technology in a constructive direction [50,51].

## 6. Conclusions

This study explored how machine learning technologies can improve healthcare operations management. In particular, this study developed and applied an AI solution for malaria diagnosis using the CNN algorithm. The findings of this empirical study suggest the effectiveness of the CNN diagnosis solution to a large number of malaria cases with a reliable accuracy of 97.81%. From a healthcare operations management perspective, these results confirmed the potential of ML-based automated diagnostic systems as an alternative to address the variance in diagnostic quality and the high cost of manual diagnostic systems that rely on direct human observation. Future studies may extend this benchmarking research framework and propositions for leveraging machine learning diagnostic solutions to improve healthcare operations and prepare innovative changes to enhance the safety and welfare of humanity in this turbulent world.

## Figures and Tables

**Figure 1 healthcare-11-01779-f001:**
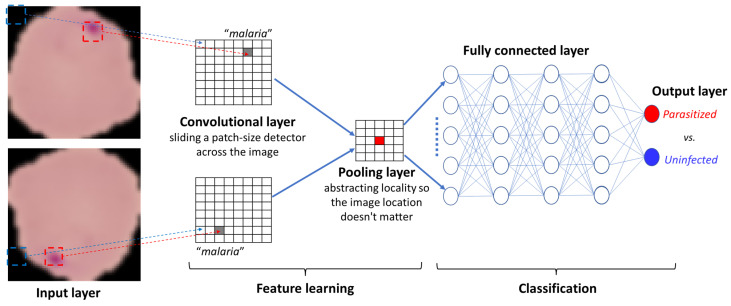
The basic architecture of convolutional neural networks.

**Figure 2 healthcare-11-01779-f002:**
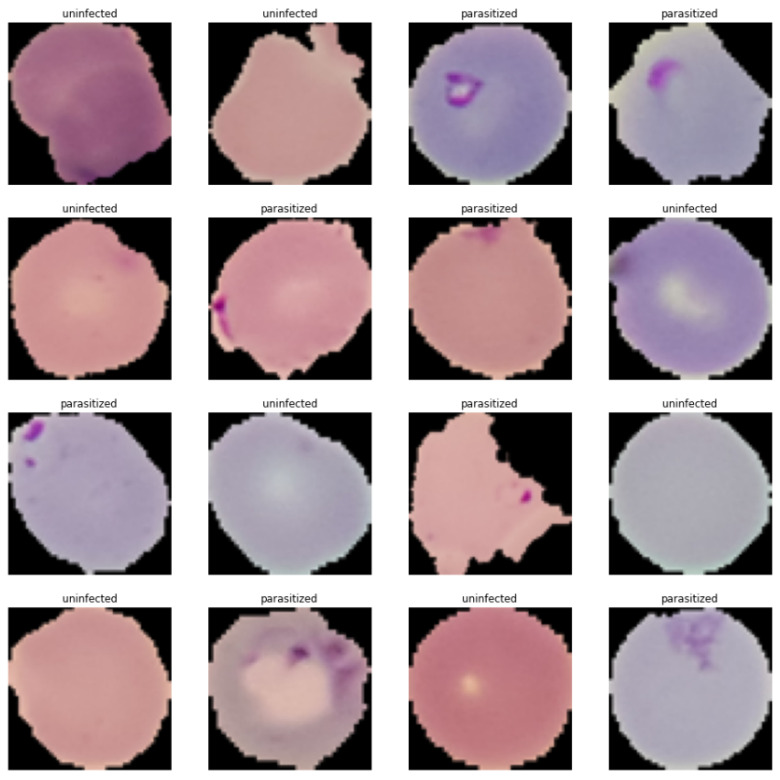
Visualized images of training data.

**Figure 3 healthcare-11-01779-f003:**
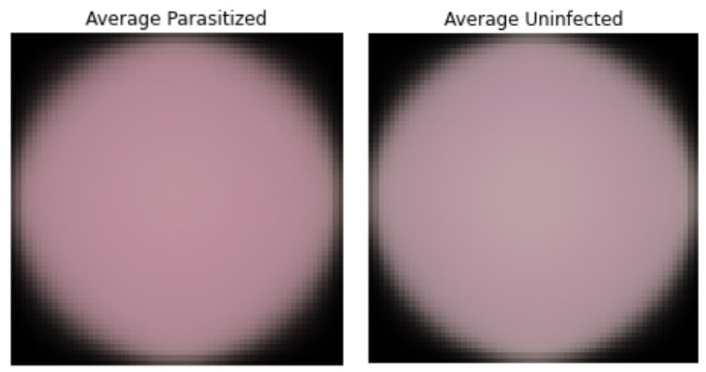
Mean images for parasitized and uninfected erythrocytes.

**Figure 4 healthcare-11-01779-f004:**
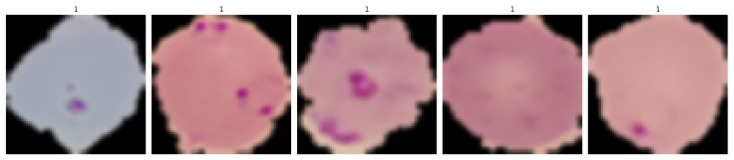
Gaussian blurring on training data.

**Figure 5 healthcare-11-01779-f005:**
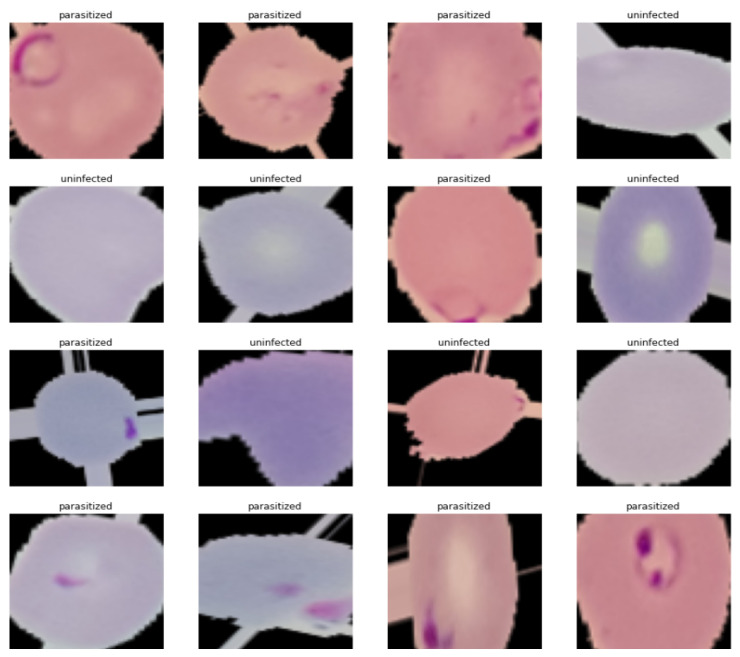
Augmented image visualization of erythrocytes in Model 3.

**Figure 6 healthcare-11-01779-f006:**
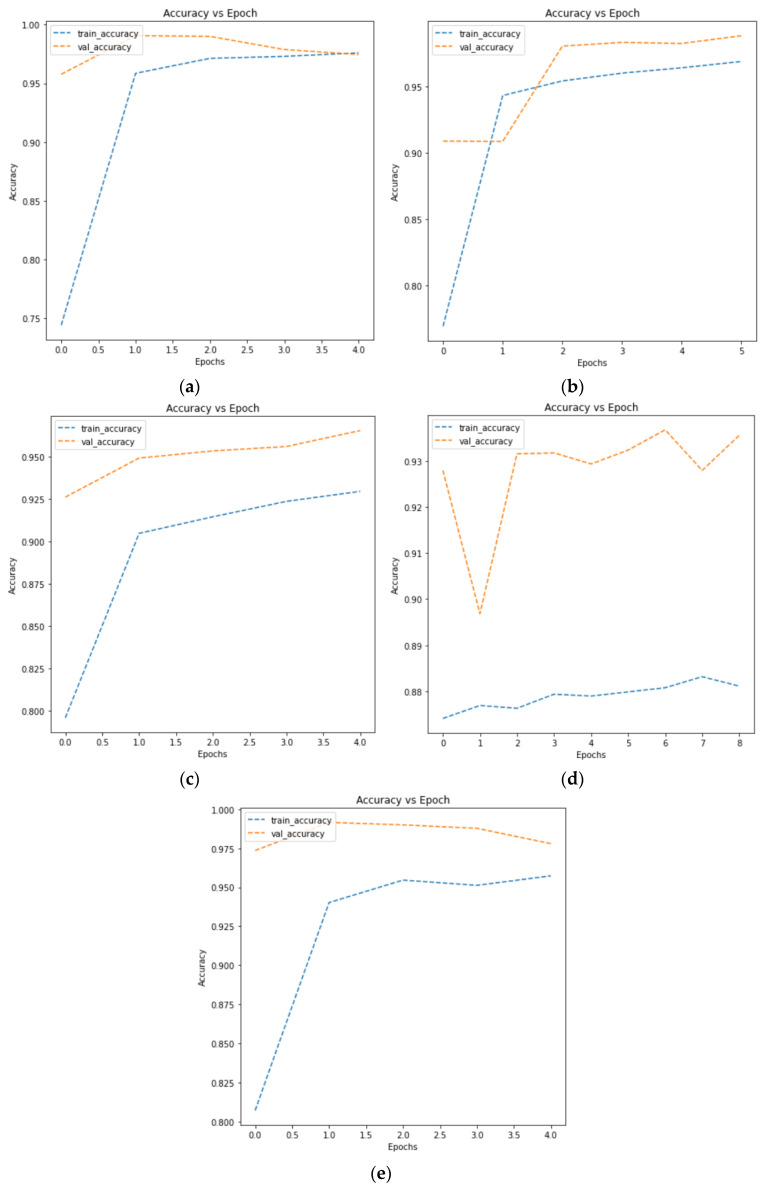
Accuracy plots of (**a**) Model 1, (**b**) Model 2, (**c**) Model 3, (**d**) Model 4, and (**e**) Model 5.

**Figure 7 healthcare-11-01779-f007:**
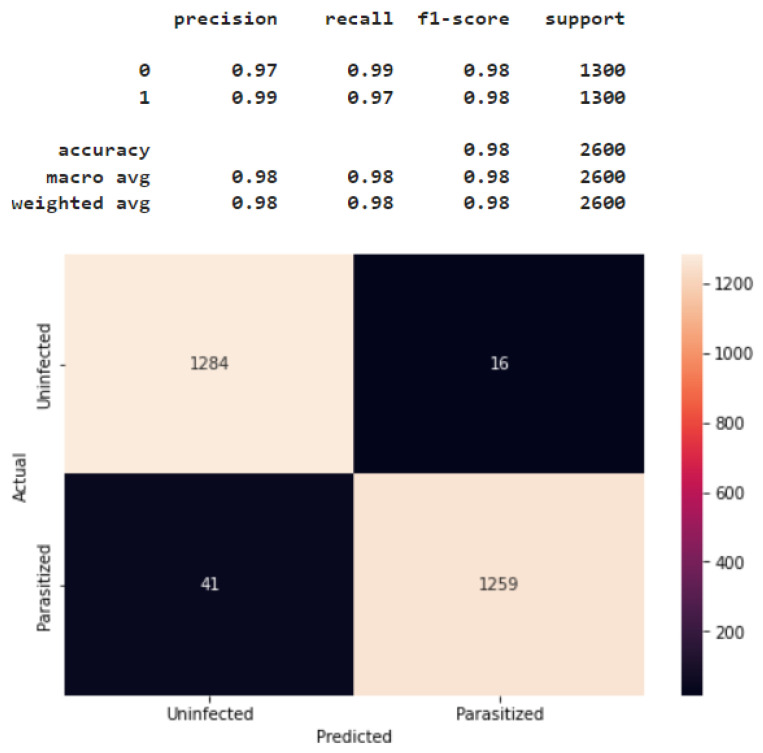
Confusion matrix for Model 2.

**Figure 8 healthcare-11-01779-f008:**
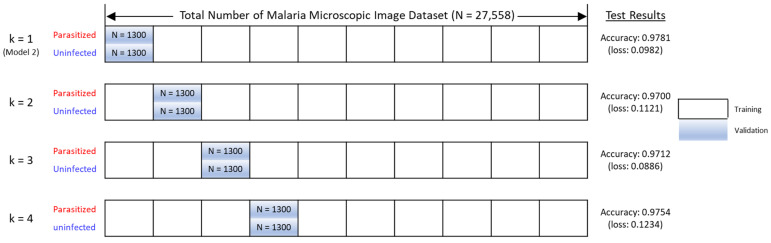
The k-fold cross-validation test.

**Figure 9 healthcare-11-01779-f009:**
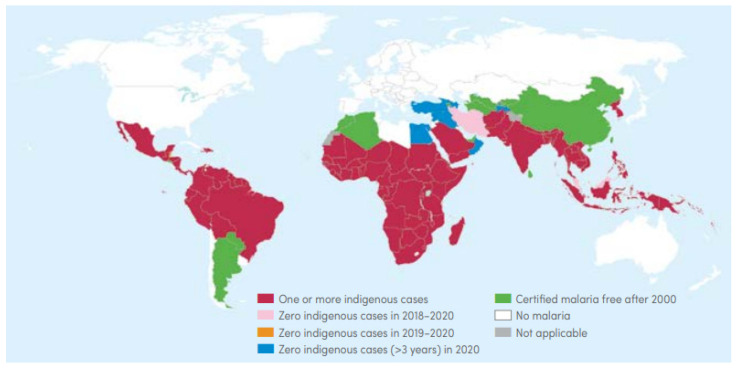
Malaria endemic countries from 2000 to 2020 (source: WHO [2], p. 79).

**Figure 10 healthcare-11-01779-f010:**
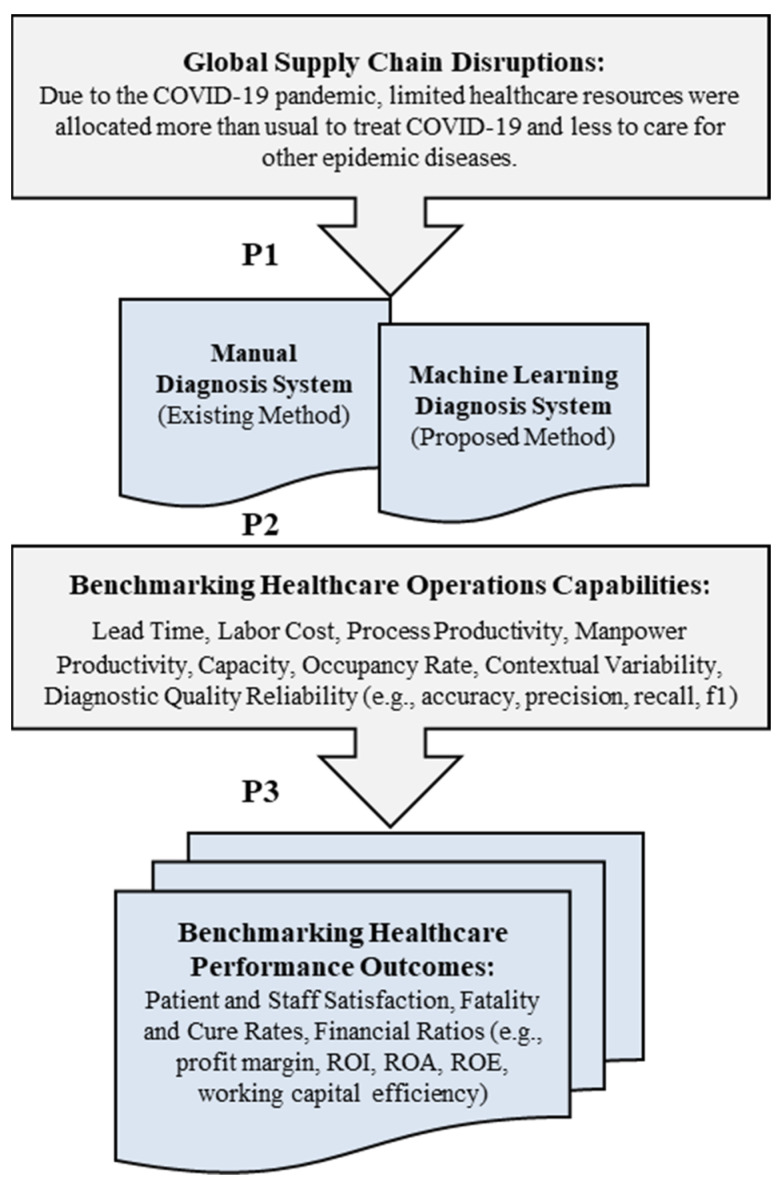
A proposed research framework for leveraging ML-based diagnosis systems to improve healthcare operational capabilities.

**Table 1 healthcare-11-01779-t001:** Comparison of techniques and their performances.

	Purpose	Accuracy
Model 1	A second convolutional layer with 64 filters was added to the base model.	0.9777 (loss: 0.0756)
Model 2	Batch normalization was performed, and a second convolutional layer with 64 filters was added to the base model.	0.9781 (loss: 0.0982)
Model 3	It was evaluated whether the performance of the model could be improved by changing other parameters in the ImageDataGenerator (i.e., Data Augmentation)	0.5000 (loss: 0.6975)
Model 4	A pre-trained mode (VGG16) was adopted to check how it performed on the model.	0.9338 (loss: 0.1874)
Model 5	It was simply tested by increasing the dropout of model 2 to 0.5.	0.9585 (loss: 0.1695)

**Table 2 healthcare-11-01779-t002:** Comparison of k-fold cross-validation results.

	Accuracy Plots	Confusion Matrix
k = 2	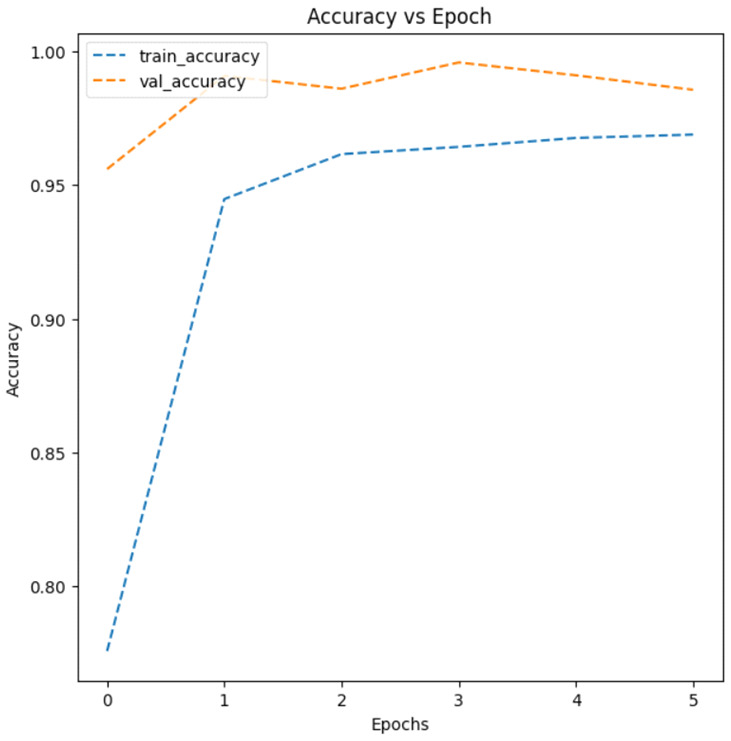	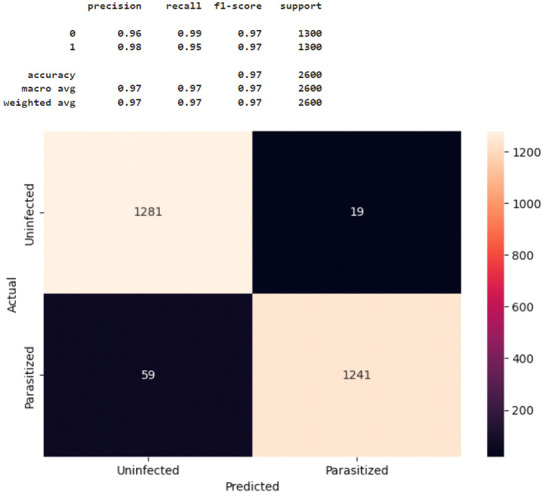
k = 3	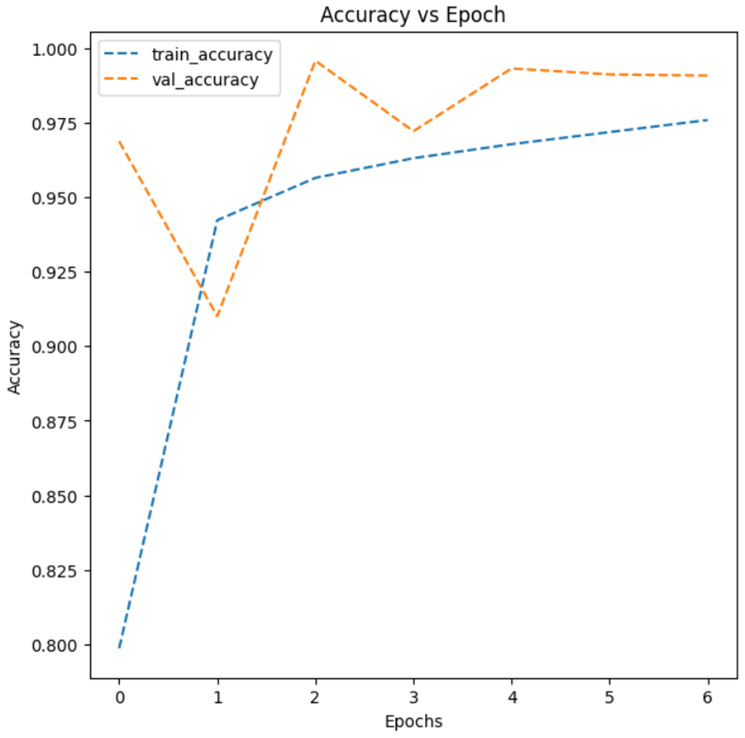	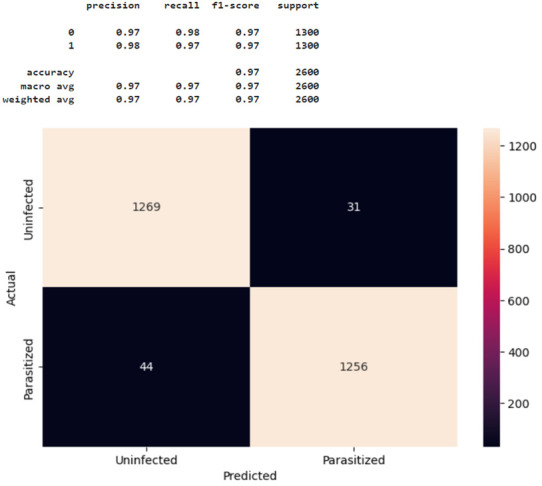
k = 4	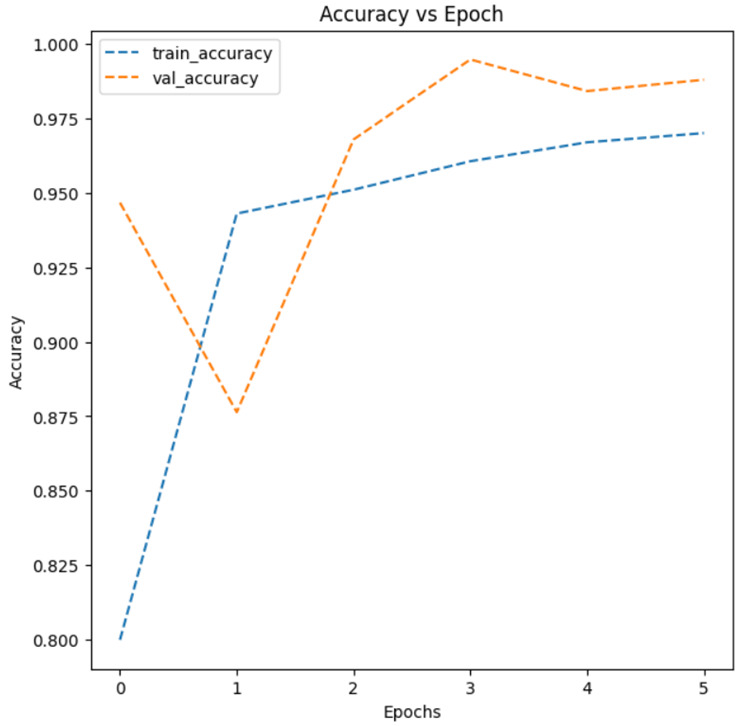	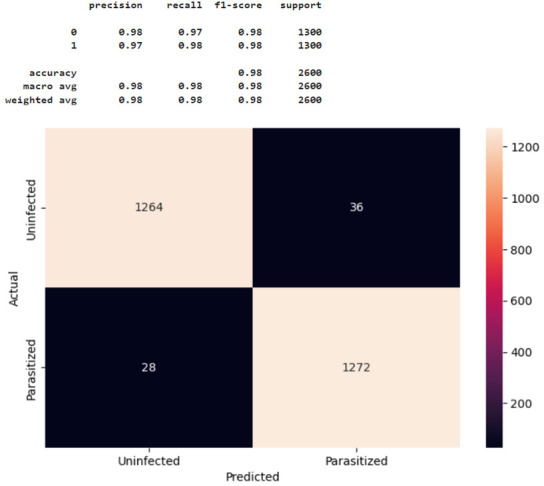

## Data Availability

The data used in this study are open source and published in the National Library of Medicine at the National Institutes of Health.

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
