# Peer review of "Applying Machine Learning to Healthcare Operations Management: CNN-Based Model for Malaria Diagnosis"

_healthcare, 2023, doi:10.3390/healthcare11121779_

Round 1
Reviewer 1 Report
The “Machine Learning and Healthcare Operations: An Empirical Study for Epidemic Diagnosis” manuscript describes the application of CNN to predict malaria from red blood cells. This paper has high public health interest and provides a novel and more efficient diagnostic methodology. The paper is well-written and clearly laid out. A minor suggestion is to include some irrelevant images as background noise in the input data to evaluate the performance of the model.
Author Response
We sincerely appreciate the comments that are very helpful in improving this paper. Point-by-point responses to specific comments are provided as an attached file. The changes made in the manuscript are highlighted in yellow.

Reviewer 2 Report
· Title of the study is misleading. It doesn’t clearly reflect the work.
· Is it appropriate to call CNN a Machine Learning based AI solution?
· Line 61 and 62 –Is it called as different architecture? It’s completely incorrect.
· Related works – While there is numerous research carried out around malaria classification and to be more specific NHI dataset. It is a popular dataset among the research community and several papers are published using the malaria dataset. But the related works focus primarily on the other types of infectious disease with more emphasis on the CNN. The related works need to be completely rewritten.
· Section 5.2 does not blend well with the study.
· Overall, the study is mere application of CNN.
Author Response

(The authors gave the same response as above.)

Reviewer 3 Report
Very good work and interesting/important results. However, there are few issues:
1) The performance is suspectedly good (~99%!), we see in figures 6-10 that the validation accuracy is higher than train accuracy. This is really unlikely. There are some possibilities that make me think of overfitting. Either you didn't run the model long enough (more epochs) or youhad some bias in selecting the validation set as the model's generalization is too good to be true and the size of the validation set is small (10% of the data) or you have some dropout/hyperparameter that is not well tuned etc. Overall, overfitting is a concern. I would check this and make sure the gap between validation and train set is correct (really less likely), you can try several iterations of train-test split and report the results on several runs if possible. I would shuffle the data before splitting to remove any bias on the validation set.
2) The figure 1 needs major improvement, it doesn't show any detail about the CNN's architecture.
3) figures 6-10 lack the quality and they can be merged in 1 figure.
4) Line 203, there is no need to bring the definition of recall/precision etc. as these are common knowledge.
5)Figure 11 is extra and should be deleted
6) Appendix C is absolutely unnecessary and should be deleted.
8) Line 104, you mention that reference 14 is an excellent source to learn deep learning. I believe they are way better sources that should be mentioned here. even though [14] is a pretty good one. Overall, in my opinion you can improve the literature review.
Author Response

(The authors gave the same response as above.)

Round 2
Reviewer 2 Report
Authors have reworked the manuscript based on the comments.
Author Response
Thank you for these positive comments about our revision! We appreciate your review and constructive feedback that could make us improve our manuscript during the revision process!
